# Primary care doctor and nurse consultations among people who live in slums: a retrospective, cross-sectional survey in four countries

Improving Health in Slums Collaborative

**Correspondence to**
Dr Samuel I Watson;
s.i.watson@bham.ac.uk

## ABSTRACT

**Objectives** To survey on the availability and use of primary care services in slum populations.

**Design** Retrospective, cross-sectional, household, individual and healthcare provider surveys.

**Setting** Seven slum sites in four countries (Nigeria, Kenya, Pakistan and Bangladesh).

**Participants** Residents of slums and informal settlements.

**Primary and secondary outcome measures** Primary care consultation rates by type of provider and facility.

**Results** We completed 7692 household, 7451 individual adult and 2633 individual child surveys across seven sites. The majority of consultations were to doctors/nurses (in clinics or hospitals) and pharmacies rather than single-handed providers or traditional healers. Consultation rates with a doctor or nurse varied from 0.2 to 1.5 visits per person-year, which was higher than visit rates to any other type of provider in all sites except Bangladesh, where pharmacies predominated. Approximately half the doctor/nurse visits were in hospital outpatient departments and most of the remainder were to clinics. Over 90% of visits across all sites were for acute symptoms rather than chronic disease. Median travel times were between 15 and 45 min and the median cost per visit was between 2% and 10% of a household's monthly total expenditure. Medicines comprised most of the cost. More respondents reported proximity (54%–78%) and service quality (31%–95%) being a reason for choosing a provider than fees (23%–43%). Demand was relatively inelastic with respect to both price of consultation and travel time.

**Conclusions** People in slums tend to live sufficiently close to formal doctor/nurse facilities for their health-seeking behaviour to be influenced by preference for provider type over distance and cost. However, costs, especially for medicines are high in relation to income and use rates remain significantly below those of high-income countries.

## INTRODUCTION

Strengthening primary care is recognised as the most 'inclusive, effective and efficient approach' to improving population health and well-being and as being key to achieving universal healthcare globally as codified in recent international declarations.[1] However, recent high-profile analyses have shown that

### Strength and limitations of this study

► This study provides the most comprehensive evidence across multiple nations to date on the facilities available to people who live in slums in general and on doctor and nurse consultations.

► We surveyed over 7000 individuals across seven slum sites in four counties on their access to and use of healthcare.

► We estimated consultation rates and provider types for primary care visits as well as the elasticity of demand for provider type with respect to time and cost of a visit.

► While several attempts were made to contact participants, our response rates ranged from 68% to 94% across the sites, and so our sample may miss some of the most vulnerable residents of the study areas.

► We could not make a reliable assessment of patient need for healthcare so it is difficult to interpret these results in terms of the equity of access to care.

little systematic evidence has been collected on services that are available, how often they are used, how much they cost and what type of service patients will choose under various constraints.[2–5] It is widely believed though that people in low-income and middle-income countries (LMICs) have poor access to high-quality primary care services.[2 3 6–8]

In this paper, we focus on a large and vulnerable group of people; those who live in slums. A recent *Lancet* series documented the many barriers that people in slums encounter in accessing services.[4 5] Following further review of the literature,[9] most evidence relates to specific programmes, particularly those concerned with vaccination and childbirth.[10] The little available evidence suggests that in many urban slums, pharmacies and drug sellers are frequently the first and only point of contact with a health system.[11 12] However, consultation with a doctor or nurse is an essential component of primary care providing diagnosis, treatment, advice and

referral. Outpatient doctor and nurse consultations can be provided by public and private facilities that range from 'single-handed', sometimes informal, practitioners in solo practices to large hospital outpatient departments.

The purpose of our study is to start to fill the above gaps in the literature, examining outpatient care services in seven slum sites across four countries: Nigeria, Kenya, Pakistan and Bangladesh. We compare visit rates to all providers that might offer primary care services including traditional/faith healers and pharmacies/medicine sellers. We then describe doctor and nurse services in more detail. Here, we aim to find out use rates of different facilities that provide outpatient doctor/nurse consultations for people who live in slums, including solo providers, clinics, hospitals and public versus private provision. We also aim to estimate how individuals trade-off characteristics like cost and time when choosing between types of provider of outpatient care.

## METHODS
### Setting
Our aim was to examine the use of outpatient consultations with a doctor or nurse. The results reported in this article form part of a broader programme of work on health and healthcare in informal settlements described in detail elsewhere.[13]

We conducted a series of household, individual and healthcare facility surveys across seven informal settlements in four countries: Nigeria, Kenya, Pakistan and Bangladesh. We selected sites that (1) fulfilled the UN definition of a 'slum'[14] and (2) were named, geographically distinguishable, neighbourhoods within city boundaries. This also provided a geographical basis on which to survey available health services.

### Study design and population
#### Household and individual surveys
The design and methods of the survey components of the Improving Health in Slums project have been published elsewhere.[13] Briefly, we aimed to conduct a spatially referenced, household-based, retrospective, cross-sectional survey in seven sites, which are described in table 1. Key indicators for the health systems of the respective countries are reported in online supplemental table C1.

All structures in each site were first mapped using satellite imagery. The resulting maps were then 'ground-truthed' through participatory geospatial mapping and corrected as required. During this stage all households in each structure were identified, which formed the sampling frame for the household survey. We used an inhibitory sampling design with close pairs to generate a spatially regular and well-dispersed sample of 1200 households for each study site,[15] assuming a response rate of 80%. Within each household, all residents were identified and an adult over 18 and a child under 12 (should one be resident) were selected at random for the individual surveys. Adult women were oversampled compared with men at a ratio of 2:1 to achieve reasonable precision in both groups as we expected greater healthcare use among women. Up to three attempts were made to

| | | **Table 1** Summary of study sites | | |

| Site | Location | Approximate population (000s)* | Approximate density (000s/km²) | Description of population and area |
|------|----------|-------------------------------|-------------------------------|-----------------------------------|
| NG1 | Ibadan, Nigeria | 5.8 | 5 | Resettled, mixed Yoruba and Hausa community at the edge of the city including a large proportion of recent migrants from the North. Structures are mostly permanent and well-spaced. Variable access to energy with poor sanitation. |
| NG2 | Ibadan, Nigeria | 5.5 | 14 | Mostly Yoruba population inhabiting a central, historical part of the city. Building mostly permanent but dilapidated with access to energy but little sanitation. |
| NG3 | Lagos, Nigeria | 8.1 | 11 | Mixed Yoruba, Ilajes and other ethnic group population in low-paying or no employment. High crime area with little development and temporary structures. Little access to basic services. |
| KE1 | Nairobi, Kenya | 24.4 | 52 | Mixed but segregated ethnicity community of generally long-term multigenerational residents. Structures are temporary with little to no access to basic services. |
| KE2 | Nairobi, Kenya | 44.9 | 83 | Mixed community of predominantly economic migrants at the edge of an industrial area. Structures are temporary with little to no access to basic services. |
| PK1 | Karachi, Pakistan | 33.5 | 91 | Mixed ethnicity and religion, mostly permanent, population working in blue collar jobs. Structures are permanent and multistory with access to transient energy and sanitation services. |
| BD1 | Dhaka, Bangladesh | 60.0 | 171 | Mostly Bengali, Muslim population working in manual services like rickshaw pulling and house work. Semi-permanent residential structures with variable access to water, sanitation and other services. |

*Estimated from data collected in this study.

complete each survey should the participant not be available at the initial attempt. For each survey instrument we sought consent from the respondent. They were provided with an oral and/or written explanation of the study and their involvement as required and then asked to provide written consent on both a paper and digital copy of the consent form. Due to an error with the sampling process, a follow-up telephone survey was conducted in PK1 to supplement the individual surveys with additional female respondents, which is described in online supplemental section B2.

## Procedures

Three instruments were used in the household surveys: (i) a household level survey of demographic and socio-economic characteristics, including monthly household expenditures across different categories; (ii) an individual adult survey enquiring about healthcare need, access and use, and health and well-being and (iii) a child (under 12 years of age) survey asking a caregiver about healthcare needs, access and use. Questions were adapted from similar studies to facilitate comparability.[16] The adult and child surveys both asked the respondents to provide details of the last time they used healthcare (if they had done so in the previous 12 months), including facility type (public clinic, private hospital, etc), who they saw (doctor, nurse, etc), the reason for the visit, the cost and time taken for the visit and questions regarding their satisfaction with the care provided. We included 'doctor's office' or 'chamber' among the list of responses to facility-type, which generally refers to a solo clinician working alone in a private office—we refer to this category as 'single-handed'. Survey instruments were translated using an iterative process involving forward and independent backward translations (survey forms are in the online supplemental adult.pdf, child.pdf and household.pdf).

## Outcomes and statistical analysis
### Summary statistics

From the individual survey data we identified all reported outpatient consultations to any type of provider and calculated visit rates per patient-year for each type of provider (doctor/nurse, pharmacy, traditional/faith, other) for new and existing conditions and for adults and children under 12 for comparison. We report crude rates as well as age-standardised consultation rates calculated with respect to both the WHO reference population,[17] and INDEPTH population.[18] For doctor and nurse visits, we further examined the proportion of visits by facility type (single-handed, public/private hospital or public/private clinic) and reported provider type (doctor or nurse).

We estimated key characteristics of these visits including the median (IQR) time required for the visit, median (IQR) cost of the visit, proportion of households spending >10% of their monthly expenditure on a visit (as used by the WHO's Global Health Observatory), reason for choosing the provider and satisfaction by site and facility type.

### Choice model

To estimate the role of price and travel time in the choice over a provider for a doctor or nurse for an outpatient consultation, we estimated a 'choice model'. We specified a random parameters logit model,[19] which we describe in detail in the online supplemental data. In the model, each option from a set of choices has an observed component (price and time), the utility of which is determined by observed (eg, age, sex) and unobserved (eg, taste, quality) characteristics. There are several examples of choice modelling for healthcare providers in LMICs.[20] The 'choice set' we examined was: private clinic, public clinic, private hospital or public hospital. We also included 'single-handed' doctor's office for Bangladesh and Pakistan, since this type of consultation was rare or non-existent in our African sites.

We studied each site separately. There were four estimates of interest from each model: the predicted proportion of visits to each provider type 'holding fixed' price and travel time, the price elasticity of demand, the travel time elasticity of demand and the average change in price (willingness to pay (WTP)) to make an individual indifferent between two options, one of which is 15 min further travel away than the other. The elasticities and WTP were calculated separately for households whose monthly consumption expenditure was above and below (International dollars) Int\$ 100 per person per month. An elasticity is interpreted as the percentage change in demand you would expect for a 1% increase in price or travel time. We allowed preferences to vary by age, sex, secondary education, seeking care for an acute or communicable condition or for chronic or generalised pain and monthly consumption expenditure. The 'price' of a visit included the consultation fee plus the travel cost, which would be known up front to the individual, and not drugs or tests, which would not be known in advance. The prices and times of travel for providers not visited were imputed based on the above-listed individual covariates.

## Patient and public involvement

Mapping of the study sites, identification of healthcare facilities and enumeration of resident households was conducted using a participatory process involving local residents. Healthcare facility managers and owners were consulted about identification of their facilities. The public were not involved in the design of the survey questionnaires, however feedback was sought from residents in a pilot survey in all sites to assess the time burden of participating. Patient and public focus groups were established to present the findings, receive feedback and provide contextualising interpretation of the results.

## RESULTS
### Household and individual sample

Overall, 7692 households participated in the surveys with 7451 individual adults and 2633 individual child surveys completed. The median response rate was 69%, varying

by site from 57% in site KE2 to 94% in site BD1. Table 2 reports demographic and socioeconomic statistics of the population-weighted sample of individual respondents by site, and for those reporting an outpatient consultation (population summaries are reported in online supplemental table C2).

### Outpatient consultation rates across providers of all types

Figure 1 shows the visit rates per patient-year to different types of provider. Between 29% (BD1) and 61% (KE1) of visits were to doctors and nurses, while the majority of other visits were to a pharmacy. In only two sites (BD1 and KE2) were pharmacies visited more frequently than doctors or nurses. The proportion of outpatient consultations that were for new conditions ranged from 61% (NG2) to 84% (KE2). Proportionately very few healthcare visits were made to traditional or faith healers.

### Outpatient consultation rates to a doctor or nurse

Table 3 reports the outpatient consultation rates by study site for both adults and children (under 12 and under 5)—equivalent rates for new conditions only are reported in online supplemental table C3. Nigerian sites had consistently the lowest outpatient consultation rates, which were comparable for adults and children: approximately 0.2–0.4 outpatient visits to a doctor or nurse for a new condition per patient-year. Rates were higher in other sites, ranging, for adults, from approximately 0.8 (PK1) to 1.5 (BD1) visits per patient-year. Apart from Nigeria, outpatient consultation rates were higher for children than adults.

### Doctor and nurse consultation rates by provider type

Figure 2 shows the proportion of outpatient consultations for a new condition by facility type. There were differences between the sites and countries. Single-handed facilities accounted for approximately 25% of adult visits in Bangladesh and Pakistan, and 50% and 20% of child visits in these countries, respectively. However, almost no visits to single-handed facilities were recorded in Nigeria and Kenya. Hospital outpatient departments and clinics accounted for comparable shares of outpatient consultations; in particular, for adults, hospital visit shares were 51% (NG1), 66% (NG2), 69% (NG3), 38% (KE1), 39% (KE2), 37% (PK1) and 21% (BD1). These figures were similar for children. Figure 3 shows the proportion of visits by provider (doctor or nurse). For the Pakistani and Bangladeshi communities, almost all outpatient consultations were with a doctor, whereas in Kenya and Nigeria a significant minority of consultations were with a nurse for both adults and children.

### Choice of providers of doctor and nurse consultations

Table 4 reports travel times, waiting times and travel, drug, tests and other costs for the different types of facility for doctor and nurse consultations. Within each site the travel time to reach each type of facility was broadly similar with median travel times generally ranging from 15 to 30 min for all types of facility (see also online supplemental figure C1).

Bangladesh and Nigeria were the most expensive locations to seek treatment, both in relative and absolute terms, with median spending ranging from Int\$ 21 to 82 depending on facility type. Median spending in Kenyan facilities ranged from Int\$ 6 to 15, and in Pakistan Int\$ 16 to 42. Medication costs accounted the bulk of the cost of an outpatient consultation in all sites: the median proportion of the total cost of a visit accounted for by drugs was 67%–100%. Consultation fees in the Nigerian and Kenyan sites were generally under Int\$5 and often zero, whereas in Pakistan and Bangladesh they were higher (approximately Int\$5–15). The median expenditure for a visit was <10% of total monthly household expenditure for almost all types of consultation across all seven sites. However, the costs were highly skewed so that in all countries except Kenya over a third of consultations would constitute more than a third of a person's total monthly household expenditure.

Table 5 reports the results from the choice model. All elasticities were below zero, showing people prefer less costly and nearer services. However, the mean estimated elasticities were almost all between zero and minus one, with the exception of in Kenya, suggesting demand was relatively inelastic with respect to both price and time. Figure 4 compares predicted share of visits to different providers, net of costs and travel times, to actual proportions of visits: there was little qualitative difference between the two. This expressed choice is similar to survey responses; only 23%–43% of respondents reported 'low fees' being a reason they chose a provider (online supplemental table C4). Demand was generally more elastic in poorer households (table 5).

The most frequently cited reasons for choosing a particular healthcare facility and provider were proximity (54%–78% of respondents), service quality (31%–95%) and cordiality (21%–65%) (online supplemental table C4). Generally, the majority of respondents were either 'very satisfied' or 'satisfied' with their visit and rated most aspects of the visit either 'very good' or 'good', aside from waiting time where negative responses were more common. Respondents were more likely to be 'very satisfied' with care at public clinics in four of the seven sites, with comparable satisfaction with other providers in the remaining sites. Respondents were also least likely to report 'very satisfied' for single-handed facilities and private clinics. The most common medical reason for seeking care was 'communicable' or 'acute conditions' (35%–69%) with only 2%–6% of respondents reporting a chronic condition as the reason for the visit (see online supplemental figure C2 and C3).

## DISCUSSION

We found that people living in slums make use of a range of primary care providers, both public and private, and from individual clinicians to hospital outpatient

**Table 2** Summary statistics of the overall study sample (all and those reporting a doctor or nurse (D/N) consultation for a new condition visit by site

| Variable | | Nigeria | | | | | | Kenya | | | | Pakistan | | Bangladesh | |
|---|---|---|---|---|---|---|---|---|---|---|---|---|---|---|---|
| | | NG1 | | NG2 | | NG3 | | KE1 | | KE2 | | PK1 | | BD1 | |
| | | All | D/N | All | D/N | All | D/N | All | D/N | All | D/N | All | D/N | All | D/N |
| **Adults** | | | | | | | | | | | | | | | |
| N | | 1278 | 391 | 840 | 215 | 802 | 201 | 1008 | 405 | 1085 | 377 | 1112 | 498 | 990 | 257 |
| Household size | | 3.6 (1.8) | 3.8 (1.8) | 4.7 (1.4) | 4.9 (1.3) | 4.6 (1.8) | 4.7 (1.8) | 3.5 (2.1) | 3.1 (2.0) | 2.5 (1.4) | 2.4 (1.3) | 5.5 (2.6) | 5.5 (2.5) | 4.3 (1.9) | 4.4 (2.0) |
| Wealth quintile (%) | Bottom | 0 | 0 | 0 | 0 | 0 | 0 | 0 | 0 | 0 | 0 | 0 | 0 | 0 | 0 |
| | Lower | 2 | 3 | 3 | 0 | 1 | 1 | 15 | 13 | 0 | 0 | 0 | 0 | 0 | 0 |
| | Middle | 49 | 48 | 51 | 47 | 24 | 24 | 59 | 61 | 55 | 54 | 83 | 83 | 21 | 19 |
| | Upper | 48 | 49 | 46 | 53 | 75 | 75 | 25 | 26 | 44 | 45 | 15 | 15 | 78 | 77 |
| | Top | 0 | 0 | 0 | 0 | 0 | 0 | 0 | 0 | 1 | 1 | 3 | 2 | 1 | 3 |
| Monthly household expenditure (Int$) | Total | 300 (248) | 362 (250) | 342 (274) | 349 (268) | 560 (364) | 580 (363) | 232 (136) | 213 (127) | 243 (127) | 274 (152) | 1122 (631) | 1179 (667) | 463 (234) | 505 (257) |
| | Per person | 99 (86) | 108 (79) | 95 (78) | 101 (87) | 136 (101) | 142 (104) | 84 (62) | 87 (63) | 118 (81) | 130 (92) | 223 (125) | 230 (138) | 116 (58) | 126 (61) |
| Age | | 39.0 (16.7) | 39.9 (17.4) | 47.8 (16.6) | 51.3 (17.7) | 42.0 (15.0) | 43.8 (15.2) | 37.6 (13.6) | 40.1 (14.7) | 34.3 (11.1) | 34.4 (11.0) | 37.6 (12.6) | 39.2 (13.1) | 34.6 (12.3) | 36.6 (12.7) |
| Sex (% male) | | 45 | 35 | 49 | 50 | 47 | 45 | 46 | 34 | 46 | 42 | 50 | 42 | 53 | 41 |
| Education (%) | Primary/Middle | 22 | 19 | 34 | 36 | 16 | 19 | 60 | 55 | 39 | 38 | 7 | 10 | 61 | 58 |
| | Secondary | 46 | 52 | 55 | 52 | 54 | 49 | 34 | 38 | 51 | 49 | 79 | 76 | 31 | 30 |
| | Tertiary | 24 | 22 | 9 | 9 | 32 | 31 | 6 | 7 | 8 | 12 | 14 | 14 | 8 | 12 |
| **Children (under 12)** | | | | | | | | | | | | | | | |
| N | | 128 | 36 | 69 | 13 | 79 | 24 | 537 | 320 | 421 | 205 | 528 | 367 | 658 | 136 |
| Household size | | 4.9 (1.5) | 4.9 (1.4) | 4.7 (1.4) | 4.9 (1.3) | 4.9 (1.4) | 5.1 (1.4) | 5.2 (1.8) | 5.2 (1.9) | 4.4 (1.4) | 4.3 (1.4) | 7.5 (3.0) | 7.4 (2.9) | 4.9 (1.6) | 5.4 (1.8) |
| Wealth quintile (%) | Lowest | 0 | 0 | 0 | 0 | 0 | 0 | 0 | 0 | 0 | 0 | 0 | 0 | 0 | 0 |
| | Lower | 2 | 1 | 3 | 0 | 1 | 0 | 13 | 13 | 0 | 0 | 0 | 0 | 0 | 0 |
| | Middle | 48 | 62 | 51 | 47 | 25 | 37 | 60 | 56 | 55 | 55 | 84 | 85 | 25 | 20 |
| | Upper | 50 | 37 | 46 | 53 | 74 | 63 | 27 | 30 | 44 | 45 | 15 | 14 | 74 | 78 |
| | Top | 0 | 0 | 0 | 0 | 0 | 0 | 0 | 1 | 0 | 0 | 1 | 1 | 1 | 2 |
| Monthly household expenditure (Int$) | Total | 365 (315) | 359 (227) | 437 (310) | 417 (254) | 579 (336) | 626 (386) | 284 (29) | 274 (152) | 276 (132) | 286 (143) | 1304 (733) | 1324 (740) | 472 (235) | 525 (252) |
| | Per person | 74 (58) | 75 (49) | 93 (60) | 92 (59) | 123 (68) | 130 (75) | 56 (29) | 130 (92) | 69 (37) | 70 (36) | 184 (112) | 194 (117) | 100 (46) | 104 (50) |
| Age | | 6.9 (3.6) | 6.0 (3.4) | 6.7 (3.7) | 4.9 (3.3) | 6.5 (3.9) | 6.2 (3.8) | 5.9 (3.7) | 5.5 (3.8) | 5.6 (3.6) | 5.1 (3.7) | 6.0 (3.7) | 5.4 (3.4) | 5.5 (3.7) | 4.7 (3.7) |
| Sex (% male) | | 44 | 33 | 57 | 51 | 43 | 40 | 55 | 52 | 48 | 46 | 53 | 53 | 52 | 62 |

Values are mean (SD) unless otherwise stated. Int$ = International dollars

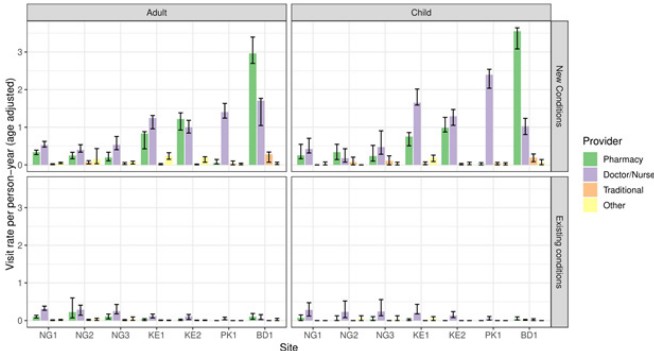

**Figure 1** Age-adjusted (to INDEPTH population) visit rates per person-year to different outpatient care providers for adults and children (under 12) for new and existing conditions.

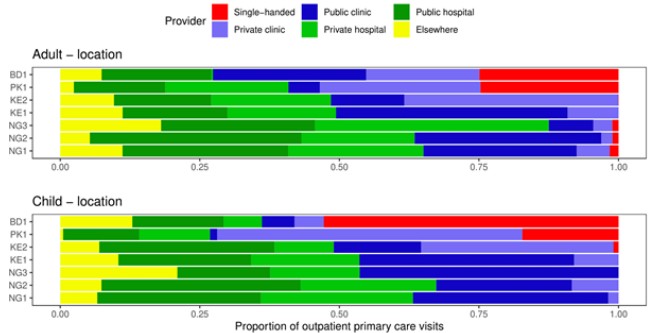

**Figure 2** Proportion of outpatient doctor and nurse consultations for a new condition by facility type.

departments. While there are considerable differences between slums and countries, we found across all sites that: traditional healers were seldom consulted for health needs, reflecting previous evidence;[21] doctors and nurses were frequently consulted, more so than pharmacies in all sites other than Bangladesh; formal clinics/hospital

outpatient departments were more popular than 'single-handed' settings; the costs of medication exceeded those of consultations and demand was relatively inelastic with respect to consultancy and travel cost.

The various providers of doctor and nurse consultations were all in relative proximity to the slum precinct; the majority of respondents reported requiring <30 min to reach their provider of choice. It is noteworthy that

**Table 3** Outpatient primary care consultation rates and outpatient doctor and nurse consultation rates (visits per person-year (95% CI))

| Site | Outpatient consultation (all conditions) | | |
|---|---|---|---|
| | Crude | WHO age adjusted | INDEPTH age adjusted |
| Adults | | | |
| NG1 | 0.42 (0.40 to 0.44) | 0.42 (0.40 to 0.44) | 0.40 (0.38 to 0.42) |
| NG2 | 0.39 (0.35 to 0.43) | 0.32 (0.30 to 0.34) | 0.29 (0.27 to 0.31) |
| NG3 | 0.28 (0.26 to 0.30) | 0.31 (0.29 to 0.33) | 0.29 (0.27 to 0.31) |
| KE1 | 1.07 (1.03 to 1.11) | 1.17 (1.13 to 1.21) | 1.04 (1.00 to 1.08) |
| KE2 | 0.93 (0.89 to 0.97) | 1.06 (1.02 to 1.10) | 0.95 (0.91 to 0.99) |
| PK1 | 0.79 (0.75 to 0.83) | 0.85 (0.81 to 0.89) | 0.77 (0.73 to 0.81) |
| BD1 | 1.52 (1.46 to 1.58) | 1.73 (1.67 to 1.79) | 1.59 (1.53 to 1.65) |
| Children (under 12) | | | |
| NG1 | 0.29 (0.19 to 0.39) | – | – |
| NG2 | 0.15 (0.05 to 0.25) | – | – |
| NG3 | 0.34 (0.20 to 0.48) | – | – |
| KE1 | 1.74 (1.62 to 1.86) | – | – |
| KE2 | 1.30 (1.18 to 1.42) | – | – |
| PK1 | 1.85 (1.73 to 1.97) | – | – |
| BD1 | 1.04 (0.96 to 1.12) | – | – |
| Children (under 5) | | | |
| NG1 | 0.30 (0.14 to 0.46) | – | – |
| NG2 | 0.21 (0.03 to 0.39) | – | – |
| NG3 | 0.40 (0.16 to 0.64) | – | – |
| KE1 | 2.57 (2.35 to 2.79) | – | – |
| KE2 | 1.68 (1.50 to 1.86) | – | – |
| PK1 | 2.46 (2.24 to 2.68) | – | – |
| BD1 | 1.50 (1.36 to 1.63) | – | – |

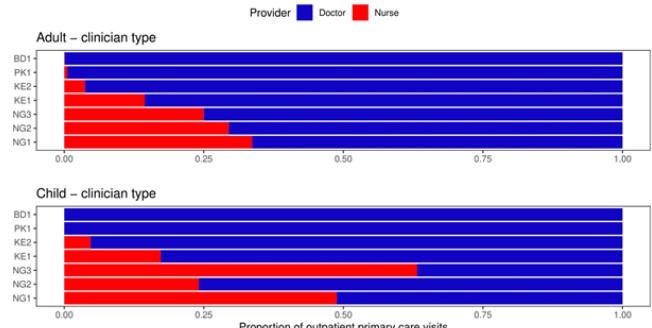

**Figure 3** Proportion of outpatient consultations for a new condition by doctor or nurse.

relatively frequent use was made of hospital outpatient departments, which accounted for as much as 50% of doctor or nurse outpatient consultations in some sites. We also found low use of individually owned and operated practices. This is in contrast to what we expected, perhaps because of the substantial literature on quality of care in 'single-handed' practices, much of it emanating from India.[3 6 7 22]

There was evidence that many households spent a significant proportion of their money on healthcare. While we found evidence that the up-front cost of a visit, including consultancy fees and travel costs, were relatively low and had only a small effect on demand for services, the total cost of a visit was much higher and often differed little between public and private facilities. Indeed, the consultancy fees even at private providers were often zero. The WHO's Global Health Observatory use the proportion of households spending >10% of their expenditure on healthcare as a marker of national health system performance and report figures of 15%, 5%, 13% and 25% for Nigeria, Kenya, Pakistan and Bangladesh, respectively (online supplemental table A1). The relative differences between countries reflect those found in this study. In both data sources, Kenya has the lowest rates of high spending relative to household spending which may be attributed to the National Hospital Insurance Fund. Nevertheless, our results suggest the proportion of slum households meeting the 10% threshold is much higher (generally >50%) than the above national averages.

Most of the cost of a primary care visit was accounted for by the cost of drugs, which reflects findings from another LMIC-based study that showed high mark-ups on medication.[23] Kenya is the only country in our study with any public funding for medication, and while respondents there did report the lowest cost to access care of any of our sites, drugs were still the biggest source of expenditure.[24] Much of the literature on healthcare access in LMICs has focused on the deleterious effect of user and consultation fees, and it has been shown that even very small costs can suppress demand and do so nonselectively and to the detriment of children.[25] However, our evidence suggests that the costs of treatment may be a larger problem for access to care in urban areas. Few visits resulted in spending on medical tests.

## Patient choice
The finding that people will frequently attend a formal facility such as a hospital outpatient department, bypassing other facilities/providers on the way, shows that people are likely willing to trade convenience and cost for perceptions of quality. We have quantified and explored these trade-offs formally by use of a 'choice model'. An immediate limitation is that we are not 'comparing like for like' since we must assume that demand for a provider type is heavily determined by the type of symptom a person is experiencing. That said, it must also be assumed that when the situation is perceived to be more serious, the more a person will eschew local in favour of more distant providers. In that sense our findings are all the more impressive, representing an underestimate of preference for formal providers when the complaint is more serious. Nevertheless, higher elasticities among the poorest people is a cause for concern.

## Overall demand for healthcare
Despite the availability of a range of providers, and study participants reporting that they were able to access healthcare when they needed it, age-adjusted use rates in our study sites were substantially lower than in high-income countries (HICs) despite a high burden of disease. In Nigeria, respondents consulted a doctor or nurse once every 3 years to once every 6–9 months in Bangladesh. Yet in the USA the crude consultation rate was 3.9 visits per person-year,[26] and in the UK it was 5.5 visits per person-year.[27] Healthcare use in our study population is lower than in HIC, and there is evidence that health needs are not being met. For instance, very few visits were reported for chronic conditions and follow-up despite the prevalence of chronic illness, which has been reported to be high in other slum populations.[28] Lack of consultation for symptoms that do not cause immediate distress may be a reason that patients presenting with cancer in LMICs are much more likely than in HICs to be in stage 3 or 4.[29 30]

## Policy recommendations
Our findings, interpreted in the context of the literature, provide the basis for an emerging policy to improve access to high-quality care in urban areas. First, while in rural areas the predominant consideration is often the existence of local services, in urban areas there are a wide variety of services in close proximity. The imperative in urban areas should therefore be to make better use of the services that already exist. For example, we have found a relatively high use of pharmacies rising to two-thirds of all healthcare contacts in Bangladesh.[11] This suggests, for example, that they might provide a good focus for dissemination of preventive health advice.

Second, our findings on expressed preference show that, in the context of the city, distance to facilities does not significantly suppress demand. These observations suggest that it might be a mistake to pursue, in urban areas, a policy to ensure yet closer location of many services to where people live. Such services would inevitably be more

**Table 4** Summary of time and costs of outpatient doctor and nurse consultations by study site for adults and children combined

| Variable | | Facility type | | | | |
|---|---|---|---|---|---|---|
| | | Single-handed | Clinics | | Hospitals | |
| | | | Private | Public/NGO | Private | Public/NGO |
| *NG1* | | | | | | |
| Travel time (min) | | – | 15 (5, 30) | 15 (10, 20) | 15 (10, 30) | 15 (10, 30) |
| Waiting time (min) | | – | 5 (2, 10) | 10 (5, 40) | 5 (2, 10) | 30 (10, 60) |
| Costs (Int$) | Consultation | – | 0 (0, 3) | 0 (0, 2) | 0 (0, 8) | 0 (0, 3) |
| | Drugs | – | 17 (4, 73) | 15 (8, 23) | 21 (10, 42) | 17 (10, 25) |
| | Tests | – | 0 (0, 7) | 1 (0, 4) | 2 (0, 8) | 2 (0, 8) |
| | Travel | – | 0 (0, 4) | 0 (0, 5) | 2 (0, 8) | 2 (0, 10) |
| | Total | – | 42 (5, 88) | 21 (13, 33) | 42 (21, 61) | 23 (14, 42) |
| Total cost as % of monthly h/h expenditure | | – | 9 (2, 21) | 7 (3, 15) | 11 (6, 19) | 9 (4, 16) |
| Households spending >10% of monthly expenditure on visit (%) | | – | 43 | 32 | 56 | 43 |
| *NG2* | | | | | | |
| Travel time (min) | | – | 20 (12, 44) | 10 (5, 20) | 15 (10, 30) | 20 (10, 30) |
| Waiting time (min) | | – | 30 (12, 55) | 15 (5, 30) | 10 (5, 30) | 30 (13, 60) |
| Costs (Int$) | Consultation | – | 0 (0, 1) | 0 (0, 1) | 0 (0, 0) | 0 (0, 4) |
| | Drugs | – | 5 (1, 11) | 15 (8, 29) | 20 (8, 42) | 17 (11, 29) |
| | Tests | – | 0 (0, 0) | 0 (0, 4) | 0 (0, 8) | 0 (0, 8) |
| | Travel | – | 0 (0, 0) | 0 (0, 4) | 0 (0, 8) | 0 (0, 8) |
| | Total | – | 13 (9, 32) | 20 (13, 34) | 29 (13, 78) | 29 (17, 42) |
| Total cost as % of monthly h/h expenditure | | – | 11 (6, 46) | 5 (3, 13) | 11 (4, 23) | 13 (5, 30) |
| Households spending >10% of monthly expenditure on visit (%) | | – | 50 | 32 | 56 | 55 |
| *NG3* | | | | | | |
| Travel time (min) | | – | 22 (10, 41) | 10 (9, 20) | 20 (14, 30) | 30 (10, 40) |
| Waiting time (min) | | – | 10 (5, 12) | 20 (5, 50) | 10 (5, 30) | 46 (10, 120) |
| Costs (Int$) | Consultation | – | 0 (0, 0) | 1 (0, 2) | 0 (0, 8) | 0 (0, 4) |
| | Drugs | – | 25 (12, 42) | 13 (8, 22) | 42 (21, 49) | 17 (8, 30) |
| | Tests | – | 0 (0, 19) | 0 (0, 4) | 0 (0, 13) | 7 (0, 17) |
| | Travel | – | 0 (0, 0) | 0 (0, 2) | 2 (0, 3) | 2 (0, 4) |
| | Total | – | 56 (29, 81) | 20 (11, 34) | 47 (33, 81) | 34 (17, 60) |
| Total cost as % of monthly h/h expenditure | | – | 10 (3, 16) | 4 (2, 9) | 9 (4, 15) | 7 (3, 12) |
| Households spending >10% of monthly expenditure on visit (%) | | – | 50 | 17 | 43 | 33 |
| *KE1* | | | | | | |
| Travel time (min) | | – | 10 (5, 20) | 15 (10, 30) | 20 (10, 30) | 28 (10, 45) |
| Waiting time (min) | | – | 30 (5, 42) | 60 (18, 94) | 20 (5, 42) | 30 (20, 120) |
| Costs (Int$) | Consultation | – | 0 (0, 2) | 0 (0, 0) | 0 (0, 2) | 0 (0, 1) |
| | Drugs | – | 9 (2, 13) | 5 (2, 9) | 7 (2, 18) | 6 (0, 9) |
| | Tests | – | 2 (0, 6) | 0 (0, 3) | 0 (0, 6) | 0 (0, 6) |
| | Travel | – | 2 (0, 5) | 0 (0, 2) | 2 (0, 3) | 1 (0, 4) |
| | Total | – | 13 (8, 30) | 6 (2, 12) | 11 (4, 30) | 8 (2, 24) |

Continued

**Table 4** Continued

| Variable | | Facility type | | | | |
|---|---|---|---|---|---|---|
| | | Single-handed | Clinics | | Hospitals | |
| | | | Private | Public/NGO | Private | Public/NGO |
| Total cost as % of monthly h/h expenditure | | – | 5 (1, 21) | 3 (1, 11) | 5 (1, 11) | 3 (1, 11) |
| Households spending >10% of monthly expenditure on visit (%) | | – | 43 | 27 | 29 | 26 |
| *KE2* | | | | | | |
| Travel time (min) | | – | 18 (10, 30) | 20 (15, 30) | 30 (15, 50) | 35 (30, 58) |
| Waiting time (min) | | – | 10 (5, 20) | 30 (10, 60) | 15 (5, 30) | 40 (20, 88) |
| Costs (Int$) | Consultation | – | 0 (0, 1) | 0 (0, 1) | 0 (0, 3) | 1 (0, 3) |
| | Drugs | – | 8 (4, 14) | 6 (2, 11) | 9 (5, 19) | 11 (7, 28) |
| | Tests | – | 3 (0, 6) | 0 (0, 2) | 9 (0, 19) | 6 (2, 11) |
| | Travel | – | 1 (0, 2) | 1 (0, 1) | 2 (1, 4) | 2 (2, 4) |
| | Total | – | 9 (4, 19) | 6 (4, 13) | 13 (6, 31) | 15 (4, 48) |
| Total cost as % of monthly h/h expenditure | | – | 3 (1, 9) | 3 (1, 8) | 4 (1, 14) | 5 (1, 15) |
| Households spending >10% of monthly expenditure on visit (%) | | – | 19 | 21 | 29 | 31 |
| *PK1* | | | | | | |
| Travel time (min) | | 10 (5, 15) | 5 (5, 10) | 20 (14, 30) | 15 (10, 30) | 30 (15, 30) |
| Waiting time (min) | | 10 (0, 25) | 10 (10, 20) | 52 (12, 98) | 25 (10, 60) | 30 (14, 60) |
| Costs (Int$) | Consultation | 3 (3, 5) | 3 (3, 6) | 0 (0, 1) | 6 (1, 29) | 0 (0, 0) |
| | Drugs | 12 (6, 16) | 9 (6, 19) | 12 (4, 16) | 16 (5, 44) | 16 (0, 31) |
| | Tests | 0 (0, 0) | 0 (0, 0) | 0 (0, 0) | 0 (0, 0) | 0 (0, 0) |
| | Travel | 0 (0, 0) | 0 (0, 0) | 1 (0, 2) | 2 (0, 6) | 2 (0, 6) |
| | Total | 18 (11, 25) | 16 (9, 24) | 17 (11, 25) | 42 (18, 104) | 19 (7, 55) |
| Total cost as % of monthly h/h expenditure | | 1 (1, 4) | 2 (1, 3) | 1 (0, 3) | 2 (4, 9) | 2 (0, 4) |
| Households spending >10% of monthly expenditure on visit (%) | | 6 | 9 | 8 | 25 | 12 |
| *BD1* | | | | | | |
| Travel time (min) | | 20 (15, 30) | 30 (20, 60) | 30 (20, 30) | 35 (19, 60) | 30 (24, 60) |
| Waiting time (min) | | 45 (30, 90) | 60 (30, 90) | 85 (30, 128) | 20 (19, 30) | 60 (30, 120) |
| Costs (Int$) | Consultation | 12 (6, 15) | 12 (1, 15) | 0 (0, 0) | 0 (0, 4) | 0 (1, 4) |
| | Drugs | 26 (15, 47) | 32 (14, 61) | 15 (5, 24) | 27 (20, 60) | 21 (9, 41) |
| | Tests | 21(8, 46) | 31 (0, 61) | 11 (1, 20) | 6 (6, 6) | 31 (9, 48) |
| | Travel | 2 (2, 3) | 2 (1, 3) | 3 (2, 5) | 2 (2, 2) | 3 (2, 6) |
| | Total | 52 (30, 89) | 82 (26, 128) | 22 (15, 39) | 36 (32, 62) | 45 (18, 96) |
| Total cost as % of monthly h/h expenditure | | 13 (6, 23) | 17 (5, 32) | 6 (2, 10) | 11 (10, 16) | 9 (3, 17) |
| Households spending >10% of monthly expenditure on visit (%) | | 58 | 67 | 28 | 75 | 44 |

All values are median (IQR) if the number of recorded visits (n) was five or more.

h/h, household; NGO, non-governmental organisation.

**Table 5** Price and travel time elasticities of demand and willingness to pay for nearer services for households by total monthly per person household consumption expenditure, values are posterior mean (95% credible intervals)

| | Price elasticity of demand | | Time elasticity of demand | | Willingness to pay for 15 min less travel time (Int$) | |
|---|---|---|---|---|---|---|
| | <Int$100 pppm | ≥Int$100 pppm | <Int$100 pppm | ≥Int$100 pppm | <Int$100 pppm | ≥Int$100 pppm |
| NG1 | −0.62 (−0.89 to −0.36) | −0.27 (−0.46 to −0.10) | −1.01 (−1.76 to −0.26) | −0.47 (−1.04 to −0.18) | 8.45 (−0.95 to 22.69) | 7.20 (1.94 to 14.51) |
| NG2 | −0.96 (−1.43 to −0.49) | −0.47 (−0.75 to −0.17) | −0.29 (−0.96 to 0.42) | −0.36 (−0.88 to −0.21) | 1.35 (−2.07 to 5.03) | 3.00 (−2.06 to 9.49) |
| NG3 | −0.45 (−1.23 to 0.23) | −0.03 (−0.53 to 0.43) | 0.00 (−0.84 to 1.02) | 0.00 (−0.61 to 0.78) | −2.39 (−39.37 to 41.45) | −2.87 (−27.97 to 33.16) |
| KE1 | −1.92 (−2.46 to −1.26) | −0.62 (−1.18 to −0.06) | −0.34 (−0.94 to 0.22) | −0.16 (−0.63 to 0.31) | 1.15 (−0.39 to 3.25) | 1.50 (−1.23 to 5.44) |
| KE2 | −1.94 (−2.44 to −1.26) | −1.49 (−1.96 to −0.89) | −0.27 (−0.87 to 0.42) | −0.27 (−0.81 to 0.40) | 1.13 (−1.08 to 3.60) | 1.03 (−0.86 to 3.10) |
| PK1 | −0.30 (−1.72 to 1.05) | −0.69 (−1.31 to −0.04) | −1.52 (−2.70 to −0.32) | −0.98 (−1.61 to −0.30) | 6.93 (2.03 to 16.04) | 6.19 (2.13 to 12.65) |
| BD1 | −0.79 (−2.38 to 0.72) | −0.28 (−1.63 to 0.76) | −1.09 (−1.88 to −0.36) | −0.21 (−0.59 to 0.17) | 12.20 (−46.93 to 72.03) | 7.03 (−18.30 to 34.48) |

dispersed into small and 'single-hande' providers where care has been shown to be of low quality (even when the practitioners are medically qualified[6]), and that training has little effect on improving care quality.[7]

Third, overall use is low while people with serious disease such as cancer and tuberculosis present late. At the supply side this should be tackled by improving care quality, such as improved diagnosis, as suggested above and also by mitigating the main cost—namely medicines. On the demand side, there is a need to continue research into barriers to appropriate health seeking for symptoms of serious disease.

**Strengths and limitations**

The response rates differed by study site. We made up to three callbacks for each sampled household to minimise selection bias, but non-responsive or non-consenting households may have differed from those that

participated, particularly as slums can be highly dynamic places. There only exist one longitudinal study of slum populations,[31] while it does not capture healthcare use statistics, it does suggest the material circumstances and levels of health spending have not varied significantly year-to-year[32] (the COVID-19 pandemic notwithstanding). Nevertheless, the findings between slums in this article are qualitatively similar in terms of behaviour and agree with other studies in this area, thus we believe these results can be used to guide policy for these populations. The age-standardised consultation rates were based on the INDEPTH and WHO reference populations[17] enabling us to make age-adjusted comparisons between study sites. However, the reference population may be out of date and may not be the most appropriate population for broader international comparisons. However, estimated rates differed little between reference populations. Clinical officers were not included in this study as a

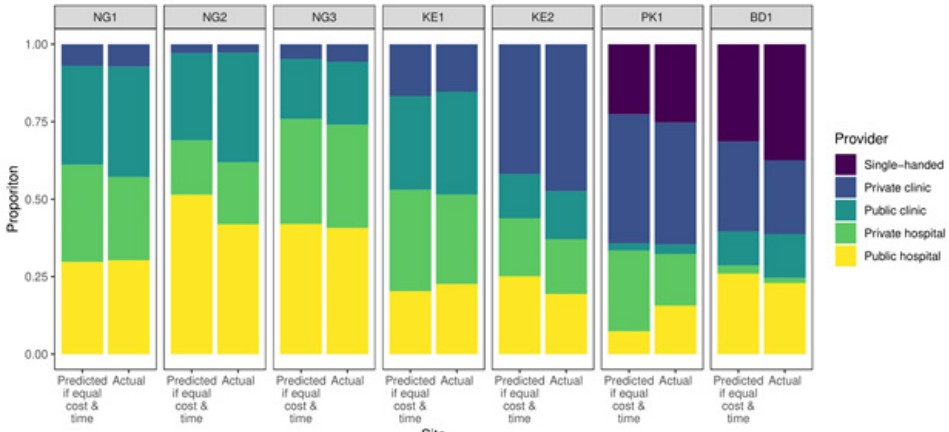

**Figure 4** Predicted proportions of visits to different provider types for an outpatient doctor and nurse consultation if price and travel time were all equivalent versus actual proportions of visits.

category, and we do not how respondents might identify or categorise them (eg, as a 'doctor' or 'other'); further research is required into the role of clinical officers in these locations. Indeed, a limitation of our study was that the type of provider (nurse or doctor) was self-reported, which means the provider could be misclassified. Further research is required to determine what effect this might have for healthcare use surveys like this. We cannot make a reliable comparison of individuals based on their health status as no reliable clinical or epidemiological indicators were captured. Differences between sites may well be attributable to large differences in population health, however we suggest that this would be unlikely. Any observed relationships, or lack thereof, may be driven by other underlying processes and confirmatory studies are needed to address some of the questions raised here. Finally, slum populations are often highly mobile, which could result in rapidly shifting population characteristics. The cross-sectional data here may therefore not provide a complete picture of slums around the world and we caution against making broad generalisations.

## CONCLUSION

There have been big 'pro-poor' improvements in key indicators, such as infant and under-five mortality, in the last 20 years. But these have been largely achieved by preventive services such as immunisation, improved nutrition and rehydration therapy. Further improvements, for example, in cancer care, will require improved clinical care along the pathways from seeking care for the first symptom to definitive treatment. We hope that our findings contribute to the debate about how to improve primary care services by supporting understanding of when, where and how the residents of poorer urban areas make contact with the health system when health needs arise.

**Collaborators** Improving Health in Slums Collaborative members (in alphabetical order): Pauline Bakibinga, Caroline Kabaria, Ziraba Kasiira, Peter Kibe, Catherine Kyobutungi, Nelson Mbaya, Blessing Mberu, Shukri Mohammed, Anne Njeri (African Population and Health Research Centre (APHRC), Nairobi, Kenya); Iqbal Azam, Romaina Iqbal, Ahsana Nazish, Narijis Rizvi (Aga Khan University, Karachi, Pakistan); Syed A. K. Shifat Ahmed, Nazratun Choudhury, Ornob Alam, Afreen Zaman Khan, Omar Rahman, Rita Yusuf (Independent University, Bangladesh, Dhaka, Bangladesh); Doyin Odubanjo (Nigerian Academy of Sciences, Lagos, Nigeria); Motunrayo Ayobola, Olufunke Fayehun, Akinyinka Omigbodun, Mary Osuh, Eme Owoaje, Olalekan Taiwo (University of Ibadan, Ibadan, Nigeria); Richard J Lilford, Jo Sartori, Samuel I Watson (University of Birmingham, Birmingham, UK); Peter J Diggle (University of Lancaster, Lancaster, UK); Navneet Aujla, Yen-Fu Chen, Christopher Conlan, Paramjit Gill, Frances Griffiths, Bronwyn Harris, Jason Madan, Helen Muir, Oyinlola Oyebode, Vangelis Pitidis, João Porto de Albuquerque, Simon Smith, Celia Taylor, Grant Tregonning, Philip Ulbrich, Olalekan A Uthman, Ria Wilson, Godwin Yeboah, Ji-Eun Park (University of Warwick, Coventry, UK); Sam Watson (Institute for Applied Health Research, University of Birmingham, Birmingham, United Kingdom).

**Contributors** Improving Health in Slums Collaborative drafted the manuscript and conducted the analyses. All members of the collaborative were involved in the design of the project, data collection, data analysis and approved the final manuscript. SIW submitted the manuscript. RJL is the guarantor.

**Funding** This research was funded by the National Institute for Health Research (NIHR) (16/136/87) using UK aid from the UK Government to support global health research. RJL is also funded from the NIHR Applied Research Collaboration (ARC) West Midlands and PG is an NIHR Senior Investigator.

**Disclaimer** The views expressed in this publication are those of the author(s) and not necessarily those of the NIHR or the UK government.

**Competing interests** None declared.

**Patient consent for publication** Not applicable.

**Ethics approval** This study involves human participants and was approved by The NIHR Global Health Research Unit on Improving Health in Slums was granted full ethical approval by the Ministry of Health, Lagos State Government (LSMH/2695/11/259), the Ministry of Health, Oyo State Government (ADB/479/657), Amref Health Africa (AMREF-ESRC P440/2018), the National Bioethics Committee Pakistan (4-87/NBC-298/18/RDC3530), the Bangladesh Medical Research Council and the University of Warwick Biomedical and Scientific Research Ethics Sub-Committee (REGO-2017-2043 AM01). Participants gave informed consent to participate in the study before taking part.

**Provenance and peer review** Not commissioned; externally peer reviewed.

**Data availability statement** Data are available upon reasonable request. The data are available from the authors on request.

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
