## [Reviewer comments · BMJ Open]

ARTICLE DETAILS

TITLE (PROVISIONAL)	Primary-care doctor and nurse consultations among people who live in slums: A retrospective, cross-sectional survey in four countries
AUTHORS	Watson, Samuel; Improving Health in Slums Collaborative, .

VERSION 1 – REVIEW

REVIEWER	Shahwar Kazmi World Health Organization
REVIEW RETURNED	22-Sep-2021

GENERAL COMMENTS	1. As per the results of the study, outpatient consultation rates were higher for children than adults, however number of individual child surveys were 1/3rd of that of individual adults surveyed, shouldn't there be more individual child surveys than individual adults? Please elaborate this or mention it as a limitation. 2. It is not clear who is a doctor, nurse or pharmacist? It would be good to define them with their minimum qualification per category in "supplementary information" section? 3. It is difficult to comprehend why profit making private clinics had "zero" consultation fee in Nigeria and Kenya? It would be good to elaborate on it. Usually, unqualified practitioners do so in LMIC and they make profit by overcharging/prescribing more drugs to compensate for their consultation fee. It would be good to know if authors tried to find out reason for zero consultation and high drug costs as per the study findings? 4. Costs for tests is almost zero at all sites except Bangladesh and KE2? Readers should know any reason behind this, please elaborate this in your discussion section. Thanks
--

REVIEWER	Leeberk Inbaraj Bangalore Baptist Hospital, Community Health
REVIEW RETURNED	08-Oct-2021

GENERAL COMMENTS	- Discussion can be strengthened by adding how the findings can help in improve primary care in this setting. - Conclusion is deviating from the findings. It has to be re-written keeping the objectives and findings in mind Overall, this a good work. I would like to congratulate the authors.
--

VERSION 1 – AUTHOR RESPONSE

Reviewer 1	
1. As per the results of the study, outpatient consultation rates were higher for children than adults, however number of individual child surveys were 1/3rd	Our child population consisted of children under 12. Not all households had children in this age range so they did not contribute an

of that of individual adults surveyed, shouldn't there be more individual child surveys than individual adults? Please elaborate this or mention it as a limitation.	individual-level child survey, hence the number of child surveys is smaller than the number of adult surveys. Our primary sampling unit was the household. We have added a sentence to explain that a child was sampled if they were resident. We do not see this as a limitation as the uncertainty due to differing sample sizes is appropriately reflected in the results.
2. It is not clear who is a doctor, nurse or pharmacist? It would be good to define them with their minimum qualification per category in "supplementary information" section?	The consultation provider was self-reported by the survey respondents. This is obviously a limitation which we have noted in the discussion.
3. It is difficult to comprehend why profit making private clinics had "zero" consultation fee in Nigeria and Kenya? It would be good to elaborate on it. Usually, unqualified practitioners do so in LMIC and they make profit by overcharging/prescribing more drugs to compensate for their consultation fee. It would be good to know if authors tried to find out reason for zero consultation and high drug costs as per the study findings?	We do not have reliable data or interviews to be able to comment definitively on this. Rather than speculate on why there was this behaviour we have just noted that this is an interesting finding requiring further research in the discussion.
4. Costs for tests is almost zero at all sites except Bangladesh and KE2? Readers should know any reason behind this, please elaborate this in your discussion section.	Note that this is patient spending on tests rather than the price charged – test spending being low (or zero) is because many patients did not undergo further medical examinations. We have added a sentence to this effect.
Reviewer 2	
- Discussion can be strengthened by adding how the findings can help in improve primary care in this setting.	We have added a sub-heading to indicate which part of the discussion our policy recommendations are in.
- Conclusion is deviating from the findings. It has to be re-written keeping the objectives and findings in mind	We are not wholly clear which part of our conclusion the reviewer objects to – we have aimed to clarify the text in terms of the objectives as requested.

VERSION 2 – REVIEW

REVIEWER	Shahwar Kazmi World Health Organization
REVIEW RETURNED	25-Oct-2021
GENERAL COMMENTS	Thanks, it is a good insightful work. Hope it will help policy makers to reshape their primary health care services for vulnerable groups as per health seeking behavior presented in this paper. Best wishes.